# Antioxidant Activity and Cytotoxicity of Aromatic Oligosulfides

**DOI:** 10.3390/molecules27123961

**Published:** 2022-06-20

**Authors:** Victoria Osipova, Yulia Gracheva, Maria Polovinkina, Daria Burmistrova, Nadezhda Berberova

**Affiliations:** 1Toxicology Research Group of Southern Scientific Centre of Russian Academy of Science, 41 Chekhova Str., 344006 Rostov-on-Don, Russia; m.hahaleva@astu.org; 2Department of Chemistry, Lomonosov Moscow State University, Leninskie gory 1-3, 119991 Moscow, Russia; jullina74@mail.ru; 3Department of Chemistry, Astrakhan State Technical University, 16 Tatisheva Str., 414056 Astrakhan, Russia; burmistrova.da@gmail.com (D.B.); nberberova@gmail.com (N.B.)

**Keywords:** aromatic oligosulfides, antioxidant, radical scavenging activity, metal chelating, cis-9-octadecenoic (oleic) acid, liver of Russian sturgeon, cancer cells, cytotoxicity

## Abstract

Natural or synthetic antioxidants with biomimetic fragments protect the functional and structural integrity of biological molecules at a minimum concentration, and may be used as potential chemotherapeutic agents. This paper is devoted to in silico and in vitro evaluation of the antioxidant and cytotoxic properties of synthetic analogues of natural compounds—aromatic oligosulfides. The antiradical and SOD-protective activity of oligosulfides was demonstrated in the reaction with O_2_^–•^ generated in enzymatic and non-enzymatic systems. It was found that phenol-containing disulfides significantly reduced the accumulation level of hydroperoxides and secondary carbonyl thiobarbituric acid reactive substances, which are primary products of oleic acid peroxidation. The antioxidant efficiency of bis(3,5-di-*tert*-butyl-4-hydroxyphenyl) disulfide increased over time due to the synergistic action of the 2,6-di-*tert*-butylphenol fragment and the disulfide linker. The highest cytotoxicity on the A-549 and HCT-116 cell lines was found for bis(3,4-dimethoxyphenyl) disulfide. Significant induction of apoptosis in HCT-116 cells in the presence of bis(3,4-dimethoxyphenyl) disulfide indicates the prospect of its use as an antitumor agent. The significant and moderate dependences revealed between various types of activities of the studied aromatic oligosulfides can be used in the development of a strategy for the synthesis and study of target-oriented compounds with predictable biological activity.

## 1. Introduction

Reactive oxygen species (ROS) are constantly produced in living organisms as a result of normal cellular metabolism. An excessive amount of ROS is produced under the influence of external factors (environmental pollution, tobacco smoke, radiation), which leads to the development of oxidative stress and subsequently to the emergence of various intractable diseases [1], among which cancer predominates [2]. Therefore, natural or synthetic antioxidants have recently been used as therapeutic agents that protect the functional and structural integrity of biological molecules at a minimum concentration [3]. Oxidative stress plays a very important role in the behavior of cancer cells. Due to the accelerated metabolism, affected cells show higher levels of ROS compared with healthy ones, and this makes them more susceptible to death [4]. The use of chemotherapy drugs that irreversibly damage tumor cells, leading to their apoptosis, is the most common method of treatment [5]. The application of available chemotherapeutic agents is often limited by the appearance of resistance, systemic toxicity, multiple side effects during prolonged use, and the absence of selective cytotoxicity for tumor cells [6]. Phytochemicals also have a number of disadvantages (such as hydrophobicity, low cellular uptake, rapid elimination) that reduce the therapeutic index and limit their application [7]. The use of additional medicament to reduce the toxicity of anticancer drugs, as a rule, leads to a decrease in the efficiency of treatment, which is an extremely undesirable process.

To overcome all unwanted effects, the application of natural products with antioxidant and antitumor properties due to the presence of biologically active compounds in their composition, including organosulfur and phenolic derivatives, is certainly preferable to synthetic analogues [8]. Biologically active components of natural plant products are regularly tested during experimental and clinical trials since they can alleviate and prevent pathological conditions [9], including cancer [10]. Various organosulfur compounds with a wide range of biological properties are found in cereals, legumes, vegetables, fruits, and other plant products [11,12]. They have been intensively tested as potential antitumor, antibacterial, cardioprotective, anti-inflammatory, anti-tuberculosis drugs, drugs against HIV, Alzheimer’s and Parkinson’s diseases, as well as antioxidants [13,14]. For example, diallyl disulphide (leaves: 34.0%; flowers: 49.7%) and diallyl trisulphide (leaves: 58.2%; flowers: 32.7%) were identified from *Adenocalymma alliaceum*, and benzylthiol (20.3%) and dibenzyl disulfide (18.0%) were detected in inflorescences of *Petiveria alliacea* [15]. These types of organosulfur compounds are also very common in garlic and onion. Garlic is one of the most common herbal products with a wide spectrum of pharmacological action and proven anticancer activity [16]. The main bioactive compound in garlic is diallyl disulfide, which has shown a cytotoxicity against breast [17], lung [18] and colon cancer cells [19], but unfortunately, its potential use is limited due to very high volatility and low bioavailability [20]. At the same time, the role of sulfides as antioxidants, including in the reduction of transition metal ions, has not been sufficiently studied in contrast to thiols [21].

Research on the synthesis and study of the properties of new diallyl disulfide derivatives, which should be more effective and safer than the natural analogue, is relevant and promising. Antitumor activity against human breast cancer cell lines [22] was investigated for a number of 4-substituted benzyl analogues of diallyl disulfide. The antiproliferative activity of diallyl disulfide was significantly increased by selecting appropriate structural fragments; disulfide with a cyano-group showed the greatest efficiency. Synthetic analogues of diallyl disulfide were obtained and their activity in vitro against human cancer cell lines was studied; bis[3-(3-fluorophenyl)prop-2-ene]disulfide demonstrated the highest activity [23]. The increased formation of intracellular ROS and cell death in the presence of this disulfide was eliminated by the addition of the known antioxidant N-acetylcysteine. This confirmed that the antiproliferative effect of bis[3-(3-fluorophenyl)prop-2-ene]disulfide is achieved through the development of oxidative stress, which triggers apoptosis.

Thus, it is currently important to synthesize new organosulfur compounds of multidirectional action with a combination of antioxidant and cytotoxic fragments in their structure, which affect only malignant cells and provide reliable protection of healthy cells. The determination of patterns in the exhibition of certain biological activity from the substance structure will help create new effective anticancer drugs that have a targeted cytotoxic effect.

Therefore, this work investigated the antioxidant properties and antiproliferative activity of synthetic analogues of natural biologically active organosulfur compounds: aromatic trisulfide **1** and disulfides **2**–**6** (Figure 1), containing an antioxidant sterically hindered phenol fragment, an S(II) atom with chelating and antiperoxide activity, a sulfide linker and/or aromatic fragment, apparently responsible for substance cytotoxicity. The conducted research should make it possible to identify the “structure-activity” relationship in studied compounds, and the obtained results will allow the targeted search for efficient and safe chemotherapeutic drugs in the future.

## 2. Results and Discussion

### 2.1. In Silico Studies

A rational approach to the development of promising new pharmacologically active compounds is based on the use of a set of research methods, including in silico prediction, which precedes in vitro and in vivo experimental studies. In this study, the forecast of biological activity of compounds **1**–**6** was performed with the help of PASS (Prediction of Activity Spectra for Substances) software [24]. The spectrum of biological activity for compounds **1**–**6** is presented in the form of a list of activity types, for which the probability of presence (P*_a_*) and the probability of lack of activity (P*_i_*) are calculated, where P*_a_* and P*_i_* values are independent and P*_a_* > P*_i_*. Among a large number of predicted types of biological activity of compounds **1**–**6**, those that are able to act as antioxidants, ROS absorbers and antidotes were selected (Table 1).

For all compounds, the probability of acting as a ROS scavenger and antidote was shown, which may be an indicator of antioxidant activity. The calculation showed that trisulfide **1** and disulfide **5** without redox active hydroxyl and methoxy groups in their structure could potentially act as ROS traps, but they could not directly act as antioxidants. The forecast of cytotoxic effect of compounds **1**–**6** in non-transformed and cancer cell lines was performed with the help of CLC-Pred (Cell Line Cytotoxicity Predictor) which is a web-service for the prediction of cytotoxicity in silico [25]. A high probability of cytotoxic action against various cancer cells, as well as against one of the most common types of cancer in women—breast cancer cells MCF-7—was predicted for all infections (Table 1).

According to the prognosis, the presence of a sterically hindered phenol group in disulfide **6** twice reduced the probability of cytotoxicity action (P*_a_* = 0.403) compared with compound **4** (P*_a_* = 0.799). These calculations were consistent with the idea that disulfide **6** exhibits more antioxidant activity (Antioxidant activity P*_a_* = 0.549) due to its structure than compound **4** (P*_a_* = 0.280).

### 2.2. Reducing Activity of Compounds

Compounds that easily release electrons or hydrogen atoms and form more stable radicals can be potential antioxidants. The redox activity of organosulfur compounds **1**–**6** can be assessed by the ability to interact with the stable radical 2,2-diphenyl-1-picrylhydrazyl (DPPH). The delocalization of the radical in the aromatic rings of DPPH ensures its high stability [27]. Therefore, this method is widely used to determine the radical scavenging activity of compounds. It is known that there are two main mechanisms by which antioxidant molecules can deactivate free radicals: hydrogen atom transfer (HAT) and single electron transfer (SET). One of them may dominate depending on the conditions, the structure of the antioxidant and the type of analysis, but both will produce identical final products, despite the difference in mechanisms [28].

It was found that only compounds **4** and **6** with phenolic groups demonstrated a radical scavenging activity (Table 2), which indicated that these compounds act mainly through the HAT mechanism; for compound **6** the IC_50_ value was 20.09 ± 0.02 μM [29].

The obtained data were consistent with the literature data, according to which sterically hindered phenols are good antioxidants due to their ability to form stable aroxyl radicals [30]. It was also previously found that cysteine, methionine, and taurine possess different activities against DPPH, and the highest one was show for the amino acid with *HS*-group [31]. The presence of two 2,6-di-*tert*-butylphenol fragments in compound **6** promoted increased antioxidant properties in comparison with the rest of the disulfides. The organic oligosulfides **1**–**3** and **5** did not exhibit antiradical activity in this reaction. Therefore, considering that this method is not universal, it is necessary to study the antioxidant activity by using other test systems.

Antioxidant activity can be evaluated by the CUPRAC test (cupric reducing antioxidant capacity) based on the ability of the compound to reduce Cu^2+^ in complex with 2,9-dimethyl-1,10-phenanthroline (neocuproin). The antioxidant capacity of compounds **1**–**6** in the CUPRAC test was measured in equivalents of the water-soluble analogue of vitamin E—Trolox (TEAC_CUPRAC_) (Table 2). The reducing ability of compounds **1**–**3** and **5** was significantly lower than Trolox, while the activity of compounds **4** and **6** was 1.8 and two times higher, respectively. The TEAC_CUPRAC_ values of disulfides **4** and **6** were quite close and indicated the manifestation of antioxidant activity mainly due to the presence of *HO-*groups. The slightly lower activity of compound **4** was explained by the absence of *tert*-butyl groups in its structure.

The assessment of antioxidant activity was also carried out by the FRAP (Ferric Reduction Aantioxidant Power) method based on electron transfer similar to the CUPRAC method [32], while in the DPPH test, the reaction proceeded according to the HAT mechanism. Antioxidant activity of compounds **1**–**6** was also calculated in Trolox equivalents (TEAC_FRAP_). All compounds demonstrated lower antioxidant activity in the reduction of Fe^3+^ to Fe^2+^ compared with the reference (Table 2). The lowest activity was obtained for trisulfide **1** and disulfide **5**, which are characterized by the absence of functional groups in the benzene ring. The TEAC_FRAP_ values for disulfide **2** with a methoxy group and compounds **4** and **6** with a phenolic fragment differed insignificantly and amounted to 0.58 and 0.69 Trolox equivalents.

Thus, the evaluation of the reducing ability of oligosulfides **1**–**6** showed that compounds with a free redox active *HO-*group could scavenge free radicals and exhibited potentially high antioxidant activity, preventing the growth of chain reactions.

### 2.3. Ferrous Ions (Fe^2+^) Chelating Activity (FIC)

Despite the important biological role of iron, it can cause various pathological diseases, such as neurodegenerative, liver and heart diseases, cancer, and diabetes. In addition, it is able to catalyze the Fenton reaction and lead to the formation of a highly reactive hydroxyl radical [33]. In this regard, the process of transition metal chelation was studied as one of the possible mechanisms of the action of antioxidants. This assay was based on the reaction of binding Fe^2+^ ions by the studied compounds under moderately acidic conditions (pH = 6). Unreacted Fe^2+^ ions reacted with ferrozine to form a stable dark violet colored complex [34]. A high absorption value corresponded to a high concentration of the formed complex of the Fe^2+^ ion with ferrozine, and as a result, a less pronounced chelating ability of the antioxidant. Complex formation does not occur in the presence of a strong chelator [35].

Organosulfur compounds **1**–**6** were able to chelate iron ions, but were significantly less effective than EDTA—a synthetic chelating agent for removing metal cations. Based on the calculated half maximal inhibitory concentrations (IC_50_, mM), it was found that diphenyl disulfide **5** (IC_50_ = 3.74 ± 0.02 mM), without any functional groups in the benzene ring, exhibited the highest chelating activity (Table 2). However, this value was an order of magnitude lower than the EDTA activity (0.38 ± 0.03 mM) [36]. Disulfides **2** and **3** with methoxy groups in the benzene ring demonstrated the least activity, while derivatives with hydroxyl groups **4** and **6** exhibited moderate iron-chelating action.

Thus, oligosulfides **1**–**6** did not exhibit pronounced Fe^2+^ chelating activity in this model system. Therefore, the studied compounds were not able to effectively inhibit metal-induced lipid peroxidation. In addition, we found no dependence of the efficiency of the oligosulfides **1**–**6** action on the number of sulfur atoms in the structure of the compounds or the presence of redox-active groups.

### 2.4. Superoxide Anion Radical Scavenging Activity

Oxygen plays a dual role in biological systems, simultaneously participating in aerobic metabolism in multicellular organisms and acting as a source of reactive oxygen species that are easily converted into toxic compounds [37]. Molecular oxygen (O_2_) reduction is an intracellular process that occurs in mitochondria under normal physiological conditions [38]. At the same time, during the reduction of molecular oxygen, compounds including free radicals, as well as active forms of nitrogen and hydrogen peroxide are formed. The superoxide radical anion (O_2_^–•^) is one of the best characterized ROS produced in vivo [39]. The superoxide radical anion is a short-lived particle and it is also capable to dismute in the reaction with water into oxygen and H_2_O_2_. In addition, O_2_^–•^ reacts very quickly with NO to form the powerful and toxic oxidizing agent peroxynitrite (ONOO-) [40].

Low levels of O_2_^–•^ regulate intracellular processes necessary for the normal functioning of the body, but in excess, O_2_^–•^ causes oxidative damage to proteins, nucleic acids and lipids, disrupting vital cellular processes and increasing mutations [41]. The use of antioxidants capable of interacting with O_2_^–•^ makes it possible to control the accumulation of ROS in the body and to prevent the induction of oxidative stress. The antiradical activity of compounds can be evaluated by their ability to utilize O_2_^–•^ obtained in various test systems. In this work, enzymatic and non-enzymatic model systems were used to generate this reactive oxygen species.

The superoxide radical anion was generated in the xanthine/xanthine oxidase enzymatic system and reduced nitroblue tetrazolium (NBT) to an intensely blue colored diformazan (*λ* = 560 nm). There was a decrease in the concentration of diformazan in the presence of a potential antioxidant that absorbs O_2_^–•^. The results of the NBT test indicated that only compounds **3** and **4** exhibited antiradical activity against O_2_^–•^ (Table 3). Despite the fact that compounds **2** and **6** were structurally similar to the corresponding disulfides **3** and **4**, they were not active in this assay.

In previous work, we found that only phenolic compounds with the *HS*-group showed 40–55% inhibitory activity [26]. Based on this fact, we suggested that the interaction with O_2_^–•^ proceeded more easily with the *HS*-group than with the phenolic *HO*-group. However, in this case, superoxide anion radical activity was found for bis(3,4-dimethoxyphenyl)disulfide **3** and 4,4’-dihydroxydiphenyldisulfide **4**, which do not contain *SH*-groups and differ in the presence of *HO*- and *CH_3_O*-groups. Thus, the different behavior of disulfides **1**–**6** in the enzymatic xanthine/xanthine oxidase system does not allow reveal of the dependence of the exhibition of antioxidant properties from the structure of the compounds.

Superoxide dismutase (SOD) is present in almost all aerobic cells and extracellular fluids and is able to inhibit the accumulation of adrenochrome by intercepting O_2_^–•^. The process of splitting O_2_^–•^ by SOD into O_2_ and H_2_O_2_ occurs in the body, which is facilitated by the presence of metal ions Cu, Zn, Mn and Fe, playing an important role in lipid peroxidation [42]. The activity of compounds **1**–**6** in the reaction with O_2_^–•^ generated in a non-enzymatic system of quinoid oxidation of adrenaline in an alkaline carbonate buffer was studied. This system is also suitable for determining the activity of the endogenous enzyme–antioxidant SOD responsible for the utilization of O_2_^–•^ [43,44]. The reaction sequence of adrenaline autoxidation with the adrenochrome formation was described earlier [45].

The value of adrenaline autoxidation in an alkaline medium without the addition of compounds **1**–**6** was taken as 100%, the calculated % inhibition indicating the antioxidant activity of the compounds. The study of the effect of oligosulfides **1**–**6** on the O_2_^–•^ formation rate in the model system of quinoid oxidation of adrenaline showed that all compounds exhibited antiradical activity (30–56% inhibition). Additionally, all compounds increased SOD protective activity of the biological product (cytosolic fraction of the Russian sturgeon liver homogenate) by 41–85%, and slowed down the adrenaline oxidation rate (Table 3). Compound **5** exhibited the highest antiradical activity under conditions of adrenaline autoxidation (56% inhibition), while compound **6** demonstrated SOD-protective activity (85% inhibition). All studied disulfides displayed better activity against O_2_^–•^ generated in the system of non-enzymatic oxidation of adrenaline in comparison with the enzymatic system xanthine/xanthine oxidase.

Thereby, we confirmed the antiradical and SOD-protective activity of disulfides **1**–**6** with respect to O_2_^–•^ generated in enzymatic and non-enzymatic model systems. The highest superoxide anion–radical scavenging activity was exhibited in the model quinoid oxidation system of adrenaline in an alkaline carbonate buffer. In this model system, there was also no unambiguous regularity in the manifestation of a greater antiradical activity with respect to O_2_^–•^ depending on the presence of a phenolic fragment or a di-/trisulfide group in the oligosulfide structure.

### 2.5. Evaluation of Lipoxygenase Inhibition

Metabolism of arachidonic acid generates many pro-inflammatory metabolites. In turn, cyclooxygenases and lipoxygenases (LOX) play an important role in the inflammatory process [46]. During oxidative stress, lipoxygenase can exhibit uncontrolled activity and cause destruction of the cell membrane due to the oxidation of phospholipids. It is known that organosulfur compounds found in plants of the onion family possess anti-inflammatory activity [47], attributing this effect to high linoleic acid composition and the ability to inhibit the expression of pro-inflammatory cytokines [48]. The experiments demonstrated the absence of anti-inflammatory activity of compounds **1**–**6,** evaluated by the ability to inhibit LOX in the corresponding assay.

### 2.6. Determination of Rate of Non-Enzymatic Peroxide Oxidation of Oleic Acid

Oxidative stress is closely associated with the development of many diseases, including cancer [49]. The formation of primary and secondary products of lipid peroxidation (LPO) are considered to be universal markers of this pathological condition [50]. Oxidation of unsaturated fatty acids by molecular oxygen, in particular oleic acid, is a good model of a peroxidation reaction in a cell membrane bilayer. During the oxidation of oleic acid, substituted radicals are formed and interact with O_2_ to form peroxyl radicals LOO^•^. The peroxyl conversion rate into the corresponding primary oxidation products (*cis*- and *trans*-isomeric hydroperoxides (LOOH)) can be used as a criterion for determining the LPO rate [51].

The total antioxidant activity of compounds **1**–**6** during the oxidation of oleic acid with atmospheric oxygen at 65 °C for 3 h was determined by standard methods according to the level of accumulation of the main metabolites. Hydroperoxides as primary products (LOOH) and secondary LPO carbonyl products formed colored complexes with thiobarbituric acid (TBARS) [52].

The kinetics of LOOH and TBARS formation was similar in the control test and in the presence of the studied compounds. The exponential kinetic curves of LOOH accumulation in oleic acid corresponded to the equation C_LOOH_ = a × e^kt^ with correlation coefficients close to 1. It indicated a pseudo-first order of reaction in the substrate, inherent in a radical process with degenerate chain branching (Figure 2). The kinetic curves of secondary carbonyl products were linear and corresponded to the equation C_TBARS_ = kt + b, with correlation coefficients close to 1 (Figure 3).

In the presence of organic sulfides, a decrease in the level of both primary and secondary LPO products was observed, and phenol-containing compounds **4** and **6** demonstrated the maximum effect. The values of relative rate constants of the LOOH formation (k_0_/k_1_) and % inhibition of the TBARS accumulation level confirmed the inhibitory effect of the studied oligosulfides (Table 3). The level of the LOOH accumulation after 3 h of an incubation of oleic acid with additives of compounds compared with the control was 46–94% (or 6–54% of inhibition), and the total content of TBARS was 17–79% (or 21–73% of inhibition).

The most effective decrease in the LOOH level was observed in the presence of compound **6** with a sterically hindered phenol moiety. Compound **4** was characterized by a greater decrease in the LOOH level than in the case of disulfide **2** with methoxy groups in the aromatic ring. Despite these data being consistent with the results of the DPPH-test, the DPPH absorption degree may not always correlate with the peroxyl radical absorption in microsomes, since LPO inhibition requires effective interaction of compounds and the cell membrane [53]. At the same time, there was no noticeable increase in the TBARS level both in the presence of compounds **4** and **6** and other oligosulfides, which confirmed their ability to decompose LOOL and LOOH without the formation of active radicals that trigger chain radical processes and contribute to the development of oxidative stress [54].

In total, the lowest antioxidant activity in this model system was noted for diphenyl trisulfide **1**. This compound inhibited the accumulation of TBARS and LOOH by 21 and 22.5%, respectively, but no inversion of properties was observed. Despite the absence of a sterically hindered phenolic fragment, compound **2** showed moderate inhibition effect (42.9%). This was consistent with previously obtained data on the ability of organosulfur compounds (in particular, di-*tert*-butyl sulfide) to act as free radical scavengers [54]. It is also known that polysulfides remain highly reactive even at elevated temperatures, which allows them to surpass phenols and alkylated diphenylamines, widely used in medicine as antioxidants, in terms of antiradical activity.

Thus, disulfides with phenolic fragments exhibited the highest antioxidant activity in the model system of oleic acid oxidation. This was explained by the intramolecular synergism of the antiradical action of two phenolic fragments and the antiperoxide effect of sulfur (II) atoms, as was shown earlier for bis-(3-(3,5-di-*tert*-butyl-4-hydroxyphenyl propyl) sulfide (thiophane) [55].

### 2.7. Determination of the Accumulation Level for TBARS in the Liver Homogenate of Russian Sturgeon

The liver is the main organ that all metabolic products pass through. In addition, proteins of the antioxidant defense system are concentrated in the liver, which prevent the toxic effects of various ROS. The oxidation of liver lipids, which compose cell membranes, directly causes cell damage in vivo. Therefore, the liver is similar to biological systems and is considered as a classic model system for studying LPO processes in vitro and in vivo [56]. We studied the antioxidant potential of the compounds on a model system of a long-term process of lipid peroxidation in sturgeon liver. The assay allowed evaluation of the effect of the compounds under prolonged oxidative stress when possible an increase in peroxidation processes and a decrease of antioxidants concentration over time, and inversion of antioxidant/prooxidant properties in vitro [57].

The level of accumulation of TBARS in the presence of compounds **2**–**4** did not significantly differ from the control test (*p* > 0.05) at all stages of LPO, which indicated the absence of pronounced anti-/prooxidant activity. The addition of diphenyltrisulfide **1** to the liver homogenate reduced the level of TBARS after 1 h of incubation by 64%, but the efficiency of the antioxidant action decreased after 24 h to 48%, and after 48 h to 37% without inversion of properties. In the previous research, we found that bis(3,5-di-*tert*-butyl-4-hydroxyphenyl) disulfide **6** exhibited a pronounced antioxidant effect at all stages of LPO, and diphenyl disulfide **5** had a prolonged prooxidant activity [29]. It was previously shown that the rate of interaction of polysulfides with peroxides was proportional to the number of sulfur atoms in the polysulfide structure [54]. However, in our study, the behavior of diphenyl trisulfide **1** and diphenyl disulfide **5** in this model system in vitro was absolutely opposite, which was presumably due to the multifactorial nature of the research system.

Is known a positive role of biologically active compounds in the prevention of human diseases, in particular flavonoids, which is associated with their direct antioxidant effect or with moderate prooxidant activity. The latter, in turn, depends mainly on the number and position of hydroxyl groups, as well as on their ability to chelate transition metal ions [58]. Some compounds induce tumor cell apoptosis by the ROS formation through a redox cycle followed by DNA fragmentation [59]. Therefore, this property is used in the development of new target compounds that can specifically affect key biotargets responsible for the vital activity of a healthy or tumor cell [60]. A number of similar drugs used in clinical practice and antitumor therapy have been developed in recent decades, for example, gold nanoparticles functionalized with the synthetic antioxidant Trolox [61].

Thus, the absence of pronounced inhibitory activity of disulfides **2**–**4** and the presence of prolonged prooxidant activity of diphenyl disulfide **5** suggest an anticancer activity for these compounds. In addition, the prooxidant effect of potential antioxidants may be an important part of the mechanism of their antitumor action.

### 2.8. Evaluation of the In Vitro Anticancer Activity

The mechanism of disulfide cytotoxicity is believed to depend on S-thiolation, which is due to the stability of the leaving group in the exchange reaction of thiolysis with proteins in the endoplasmic reticulum [62]. Ajoene ((2-propenyl-3[3-(2-propenylsulfinyl)-1-propenyl] disulfide) is a rearrangement product of the primary product of allicin isolated from garlic and exhibits cytotoxicity against cancer cells in the micromolar range [63]. Ajoene with vinyl disulfide moiety in the structure [64] is rarely found in other natural products. BisPMB (1,8-(bis-*p*-methoxyphenyl)-2,3,7-trithiaocta-4-ene-7-oxide), a synthetic analogue of ajoene [65], inhibits protein synthesis and increases the level of ubiquitinated proteins. The authors note that BisPMB exhibits selective cytotoxicity against cancer cells at micromolar concentrations.

The antiproliferative activity of aromatic oligosulfides **1**–**6** was determined against human cell lines, including colon carcinoma HCT-116, breast adenocarcinoma MCF-7, lung adenocarcinoma A-549, colorectal carcinoma SW-480, and non-tumourigenic WI-38, using the MTT assay (Table 4). In general, the compounds were found to be moderately toxic. However, some selectivity for cancer cells was evident and the IC_50_ values for non-tumourigenic cells were slightly higher for compounds **1** and **2**.

The disulfide **1** on HCT-116 and compound **3** on HCT-116 and A-549 cell lines exhibited the lowest IC_50_ values. Compound **6** containing 2,6-di-*tert*-butylphenol fragments and disulfide bridge, was studied as a comparison. According to the results, the introduction of a spatially obstructed phenolic fragment into the structure of the antioxidant led to a decrease in the toxicity of the compound as a whole. This was consistent with the data of other groups [66,67]. Comparison of compound **5** with prolonged prooxidant activity on a model system of long-term sturgeon liver LPO and diphenyl trisulfide **1** with a relatively high antioxidant activity, demonstrated that both compounds exhibited similar cytotoxicity against cancer cells. Among the oligosulfides studied, compounds **2** and **3** demonstrated the maximum cytotoxicity on HCT-116 and A-549 cancer cells. Thus, the presence of methoxy groups in these compounds increased the antiproliferative activity, opening up the possibility for their further study as cytotoxic agents with minimal side effects.

### 2.9. Apoptosis Induction and Cell Cycle Analysis

Apoptosis or programmed cell death is an important and active regulatory pathway of cell growth and proliferation. Cells respond to specific induction signals by initiating intracellular processes that result in characteristic physiological changes. Among these are externalization of phosphatidylserine (PS) to the cell surface, cleavage and degradation of specific cellular proteins, compaction and fragmentation of nuclear chromatin, and loss of membrane integrity. Phosphatidylserine (PS) is normally located in the inner part of the cell membrane. In the initial stage of apoptosis, the PS residues which translocated to the extracellular are detected by Annexin V, which is a calcium-dependent phospholipid-binding protein with a high affinity for PS. Annexin V flow cytometric assay was used to evaluate the compounds’ ability to induce apoptosis and influence on cell death. HCT-116 colon carcinoma and A-549 lung adenocarcinoma cells were treated by disulfides **2** and **3**. The compounds were added to cultured cells at twice the IC_50_ concentration and incubated for 24 and 48 h. It was found that compound **3** induced apoptosis in HCT-116 cells (Figure 4). The percentage of total apoptotic cells were 11.8 % at 38 µM (2 × IC_50_), and doxorubicin treated cells (positive control) showed 17.9% of apoptosis. The majority of apoptotic cells after treatment with **3** were observed in earlier apoptosis at 9.1%. The percentage of late apoptotic cells were 2.7% for compound **3** and less than 1% for doxorubicin.

Significant changes were observed in the apoptotic profile of compound **3** for the HCT-116 cell line after 48 h. Compound **3** intensely induced apoptosis (Figure 5). The total number of apoptotic cells increased significantly to 44%; moreover, the percentage of cells in late apoptosis also increased from 2.7 to 17.7%.

The apoptotic profile of A-549 cells showed the inducing activity of compounds **2** and **3.** The most interesting pattern was observed after 48 h, where the total number of apoptotic cells for **2** and **3** varied from 25.2 to 42.6% (Figure 6). However, compound **3** after 48 h with four methoxy groups was more toxic than **2** with two methoxy groups towards A-549 cells. The percentages of early apoptotic cells were 38.2% and 24.0%, respectively. Thus, after 24 and 48 h of incubation, HCT-116 and A-549 cells were sensitive to the action of compounds **2** and **3**. It could also be assumed that the number of methoxy groups played a significant role in the induction of apoptosis.

The effect of compounds on the cell cycle was studied by mixture of reagents including the nuclear DNA intercalating dye propidium iodide (PI) and RNases in a proprietary formulation after incubation for 24 h. PI discriminates cells at different stages of the cell cycle, based on differential DNA content in the presence of RNAse to increase the specificity of DNA staining. Resting cells (G0/G1) contain two copies of each chromosome. As cells begin cycling, they synthesize chromosomal DNA (S phase). Fluorescence intensity from PI increases until all chromosomal DNA has doubled (G2/M phase). At this stage, the G2/M cells fluoresce with twice the intensity of the G0/G1 population. The G2/M cells eventually divide into two cells. The assay utilized PI-based staining of DNA content to discriminate and measure the percentage of cells in each cell cycle phase (G0/G1, S, and G2/M). To assess the effect of the compounds on cell cycle arrest, the analysis was also performed on the HCT-116 cells. To clarify the effect of compound **3** on the cell cycle, HCT-116 cells were treated for 24 h with compound at double concentration of IC_50_. The effect of the compound **3** on the cell cycle for HCT-116 cells is presented in Figure 7. Significant changes of cell cycle distribution were registered by 24 h. We observed a sizeable increase in G2/M events concomitant with the S phase decrease.

### 2.10. Pearson’s Correlation Analysis

To assess the relationship between various indicators of the antioxidant activity of organosulfur compounds **1**–**6**, a Pearson’s correlation analysis was carried out [68], and the calculated coefficients are presented in Table 5.

Moderate positive correlations were observed between IC_50_ values for cell viability found against MCF-7, SW-480, A-549 and HCT-116 cell lines and almost all parameters of antioxidant activity in the range r = 0.5062 ÷ 0.8338. Significant positive correlations were observed between IC_50_ values for cell viability found against all cell lines (r = 0.9225 ÷ 0.9884).

Significant and moderate negative correlations were noted between IC_50_ values for cell viability found against MCF-7, SW-480, A-549 and HCT-116 cell lines and the accumulation rate of LOOH in oleic acid (r = −0.7703 ÷ −0.9099). Significant positive correlations were observed between the reducing activity of CUPRAC and iron chelating activity (FIC), and reducing activity of FRAP (r = 0.9595 and 0.9935, respectively). It was interesting to note that a positive correlation was observed between cytotoxicity of compounds and their antiradical activity against O_2_^−•^, and the negative correlation was noted between cytotoxicity and SOD-protective activity of compounds.

## 3. Materials and Methods

### 3.1. General Procedures and Syntheses

All reagents were purchased from Sigma-Aldrich unless specified otherwise. The studied disulfides **4** (TCI, 98%) and **5** (Sigma Aldrich, Burlington, MA, USA, 99%) are commercially available compounds. Disulfide **6** was synthesized according to the previously described method with slight modifications [29,69].

All experiments were performed with a 96-cell microplate spectrophotometer Multiskan Sky, Thermo Fisher Scientifics (Waltham, MA, USA). The NMR spectra were measured in CDCl_3_ on a Bruker Avance HD 400 spectrometer with a frequency of 400 MHz (^1^H) and 100 MHz (^13^C) using Me_4_Si as an internal standard. The chemical shift values were given in ppm with the reference to solvent. The elemental analysis was carried out on Euro EA 3000 (C,H) and Analytik Jena multi EA 5000 (C,S) elemental analyzers. The GC-MS was performed on a Shimadzu GCMS-QP2010 Ultra instrument equipped with mass spectrometric (EI, 70 eV) and flame photometric detectors. Column temperature was programmed as follows: *T*_0_ = 50 °C (isotherm 1 min), *T*_1_ = 200 °C (isotherm 10 min), *T*_2_ = 280 °C (isotherm 60 min), total analysis time *τ* = 82 min.

Diphenyl trisulfide **1** was prepared according to the known method [70]: thiophenol (5 mmol, 0.5 mL, 1 eq.) was added slowly over about 1 h into suspension of sulfur (1 eq.) and *n*-propylamine (0.02 eq.) in methylene chloride at room temperature. Evolution of H_2_S began immediately and continued about 4 h after the thiol addition. When H_2_S evolution ceased mixture was filtered, the solvent was removed by vacuum distillation. Compounds **2** and **3** were obtained by the standard procedure of the thiols oxidation with bromine in ethanol [71]: the bromine solution (2.5 mmol) in ethanol was added dropwise to a solution of the corresponding thiol (5 mmol) in EtOH until the reaction mixture turned a characteristic light brown color, which indicated the completion of the reaction. The formed HBr was neutralized by NaOH, the mixture was concentrated under vacuum, and the residue was recrystallized from acetonitrile. Physical–chemical characteristics were consistent with the literature data [72].

Diphenyl trisulfide (**1**). Yield 0.425 g (68%), white crystals, mp 58–60 °C (lit. 51–52 °C [72]). ^1^H NMR (400 MHz, CDCl_3_, δ, ppm): 7.50 (m, 4H, 2×C_6_H_5_), 7.28–7.21 (m, 6H, 2×C_6_H_5_); ^13^C NMR (CDCl_3_, 100 MHz, ppm): 135.05, 128.93, 127.40, 127.12. Elemental analysis calculated for C_12_H_10_S_2_ (%): C 57.56, H 4.03, S 38.42. Found (%): C 57.30, H 4.31, S 38.70. Mass spectra (EI, 70 eV), *m*/*z* (*I*(%)): 250 [M^+^] (5), 218 (100), 185 (25), 154 (35), 140 (10), 109 (80), 77 (55), 65 (30).

Bis(2-methoxyphenyl) disulfide (**2**). Yield 0.59 g (85%), pale yellow crystals, mp 120–121 °C (lit. 119 °C [73]). ^1^H NMR (400 MHz, CDCl_3_, δ, ppm): 7.63 (d, 2H, arom. 2×C_6_H_4_), 7.60 (d, 2H, arom. 2×C_6_H_4_), 7.39 (d, 2H, arom. 2×C_6_H_4_), 7.00 (d, 2H, arom. 2×C_6_H_4_), 3.80 (s, 6H, 2×CH_3_O); ^13^C NMR (CDCl_3_, 100 MHz, ppm): 158.41, 131.02, 126.61, 121.38, 117.86, 111.95, 55.43. Elemental analysis calculated for C_14_H_14_O_2_S_2_ (%): C 60.40, H 5.07, S 23.04. Found (%): C 60.20, H 5.18, S 23.25. Mass spectra (EI, 70 eV), *m*/*z* (*I*(%)): 278 [M^+^] (30), 139 (24), 109 (76), 96 (56), 77 (47), 65 (85), 45 (93).

Bis(3,4-dimethoxyphenyl) disulfide (**3**). Yield 0.74 g (88%), pale yellow crystals. ^1^H NMR (400 MHz, CDCl_3_, δ, ppm): 7.14 (d, 2H, arom. 2×C_6_H_3_), 6.86 (d, 2H, arom. 2×C_6_H_3_), 6.83 (d, 2H, arom. 2×C_6_H_3_), 3.78 (s, 6H, 2×CH_3_O), 3.75 (s, 6H, 2×CH_3_O); ^13^C NMR (CDCl_3_, 100 MHz, ppm): 152.12, 149.66, 128.78, 120.71, 116.56, 112.19, 56.07, 55.96. Elemental analysis calculated for C_16_H_18_O_4_S_2_ (%): C 56.78, H 5.36, S 18.95. Found (%): C 56.58, H 5.60, S 19.02. Mass spectra (EI, 70 eV), *m*/*z* (*I*(%)): 338 [M^+^] (26), 169 (62), 154 (35), 125 (42), 109 (27), 96 (48), 77 (29), 65 (26), 45 (38).

### 3.2. In Silico Studies

The spectrum of biological activity of compounds **1**–**6** was predicted in silico using PASS (PharmaExpert.ru ©2011–2017·Version 2.0). The structures were drawn using the CHEM Sketch package 11.0 from the ACD chem. Laboratory. The spectrum of biological activity for substances is presented in the form of a list of types of biological activity, for which the probability of presence (P*_a_*) and probability of lack of activity (P*_i_*) are calculated. P*_a_* and P*_i_* values are independent, and their values vary from 0 to 1. In this paper, the types of biological activity for which P*_a_* is more than *P_i_* were evaluated.

The forecast of cytotoxic effect of compounds **1**–**6** in non-transformed and cancer cell lines was performed with the help of CLC-Pred (Cell Line Cytotoxicity Predictor), which is a web-service for the prediction of cytotoxicity in silico [25].

### 3.3. DPPH Radical Scavenging Activity

The free radical scavenging activity was evaluated using the stable radical 2,2-diphenyl-1-picrylhydrazyl (DPPH), according to the method described by Brand-Williams [74] with a slight modification. Solutions of compounds in MeOH were studied at a concentration of 0.2 mM. The stock DPPH solution contained 0.2 mM of radical in MeOH. An amount of 0.1 mL of the test compound solution was added to 0.1 mL of DPPH solution (0.2 mM) in each cell so that the initial DPPH concentration in cells was 0.1 mM. The microplate was placed in a spectrophotometer and the decrease in the absorbance values of DPPH solution for 40 min at 20 °C was measured at λ_max_ = 517 nm. The results were expressed as scavenging activity, calculated as follows: scavenging activity, % = [(A_0_ − A_1_)/A_0_] × 100, where A_0_ is the optical density of the control solution DPPH, and A_1_ is the optical density of the reaction mixture solution.

### 3.4. Cupric Reducing Antioxidant Capacity (CUPRAC Assay)

Neocuproine (2,9-dimethyl-1,10-phenanthroline) and Trolox were used with no further purification. The method proposed by Apak et al. was used with slight modification [75]. For these measurements, 0.05 mL of CuCl_2_ solution (0.01 M), 0.05 mL of MeOH neocuproine solution (7.5 mM) and 0.05 mL of ammonium acetate buffer solution (1 M) were added to a test tube, followed by mixing with the 0.05 mL tested compounds (0.5 mM). The mixtures were kept at room temperature for 30 min. Absorbance was measured at 450 nm against a reagent blank. The results were presented in Trolox equivalents (*Trolox equivalent antioxidant capacity, TEAC*) obtained using absorbance data, and the linear calibration curve was plotted as absorbance vs. Trolox concentration.

### 3.5. Ferric Reducing Antioxidant Power (FRAP Assay)

The sample solution (0.1 mL) was mixed with 0.1 mL phosphate buffer (0.2 M, pH 6.6) and 0.1 mL potassium ferricyanide (1%). For each test compound different concentrations in EtOH were used (5, 10, 25, 50, 100, 200 and 250 μM). The resulting mixture was incubated at 50 °C for 20 min. After the incubation period, 0.1 mL trichloroacetic acid (10%), 0.5 mL deionized water, and 0.1 mL ferric chloride (0.1%) were added to the mixture. The sample absorbance was read at 700 nm with a 96-cell microplate spectrophotometer. The reduction of Fe (III) to Fe (II) could be expressed as % inhibition or equivalent of a standard compound (e.g., Trolox) [76].

### 3.6. Ferrous Ions (Fe^2+^) Chelating Activity (FIC)

The chelation of ferrous ions by compounds was estimated by method of Dinis et al. [77]. Briefly, 10 μL of 2 mM FeCl_2_ was added to 20 μL of the investigated compound (5 mM) and 150 μL of EtOH. The reaction was initiated by the addition of 0.04 mL of 5 mM ferrozine solution. The mixture was left to stand at 35 °C for 10 min. The absorbance of the solution was thereafter measured at 562 nm. The percentage inhibition of ferrozine–Fe^2+^ complex formation was calculated as [(A_0_ − A_s_)/A_s_] × 100, where A_0_ is the absorbance of the control, and A_s_ is the absorbance of the compound/standard. Na_2_EDTA was used as the positive control.

### 3.7. Inhibition of Superoxide Radical Anion Formation by Xanthine Oxidase (NBT Assay)

EDTA, xanthine, bovine serum albumin, nitroblue tetrazolium, and xanthine oxidase (25 MU) were purchased from Sigma Aldrich, Burlington, MA, USA. The superoxide anions were generated enzymatically by the xanthine oxidase system. The reaction mixture consisted of 2.70 mL of 40 mM sodium carbonate buffer containing 0.1 mM EDTA (pH 10.0), 0.06 mL of 10 mM xanthine, 0.03 mL of 0.5% bovine serum albumin, 0.03 mL of 2.5 mM nitroblue tetrazolium, and 0.06 mL of the sample solution in DMSO at the concentration of 5 mM. An amount of 0.12 mL of xanthine oxidase (0.04 units) was added to the mixture at 25 °C, and the absorbance at 560 nm (by formation of blue formazan) was measured by microplate spectrophotometer for 60 s. A control experiment was carried out by replacing the sample solution with the same amount of DMSO.

Inhibition I (%) = [(1 − A_i_/A_0_) × 100%], where A_i_ is the absorbance in the presence of the testing compound, A_0_ is the absorbance of the blank solution [78]. All experiments were performed three times.

### 3.8. SOD-Protective Activity and Pro-/Antioxidant Activity

SOD-protective activity of the biopreparation was the ability to utilize the superoxide anion radical O_2_^–•^, as determined by the method of Sirota [79]. A cytosolic fraction of the Russian sturgeon liver homogenate was used as the source of SOD. First, the sturgeon liver was washed with cold 0.2 M Tris (tris (hydroxymethyl) aminomethane) buffer (pH 7.8) to remove any traces of blood. All of the procedures were performed at a temperature of 0–4 °C. Next, a homogenate was obtained using a Potter homogenizer (Thomas Scientifc, Swedesboro, NJ, USA) in 0.2 M Tris buffer at a ratio of 1:10. The homogenate was then centrifuged for 10 min at 1000× *g* to remove partially destroyed cells and nuclei. The resulting supernatant contained the enzymes of the cytosolic fraction of the liver homogenate, including the SOD. Here, 10 µL of the biopreparation was added to a cuvette with 200 µL of bicarbonate buffer (pH 10.65), 10 µL of the tested compound (initial concentrations of these compounds 25 µM) and 10 µL of 0.1% adrenaline solution and was thoroughly and quickly mixed. The rate of adrenaline oxidation without and in the presence of the biopreparation was evaluated by the change in optical density, measured at 347 nm for 3 min. The decrease in the rate of the process in the presence of the biopreparation was used to characterize the SOD-protective activity.

### 3.9. Lipoxygenase Activity

LOX type 1-B from Glycine max (soybean), boric acid, linoleic acid, ammonium acetate, copper (II) chloride, EtOH (96%) were purchased from Sigma-Aldrich and were used with no further purification. LOX inhibition activity was determined spectrophotometrically by measuring the increase in absorbance at 234 nm for the oxidation of linoleic acid [80]. The reaction mixture contained the test compounds dissolved in DMSO at initial concentrations of 0.05 ÷ 2 mM or 0.03 mL of DMSO (blank), and 1 mL of 0.3 mM linoleic acid in borate buffer (pH 9.0) and 0.3 mL of borate buffer. The total sample volume was 1.5 mL, the final concentration in DMSO was 0.33% *v*/*v*. The reaction was started by adding 0.17 mL of lipoxygenase solution (500 units) in borate buffer. The increase in absorbance was measured every 10 s during 10 min under controlled temperature 25 °C. The degree of LOX activity (A, %) in the presence of the compounds was calculated according to the reported method [81].

A, % = (ν_0_ in the presence of inhibitor/ν_0_ in the absence of inhibitor) × 100%, where ν_0_ is the initial rate.

### 3.10. Determination of LOOH and TBARS Concentrations in Oleic Acid

The determination of oleic acid oxidation level was performed by the kinetic measurement of the total concentration of the corresponding isomeric LOOH using iodometric titration [52]. The oxidation of constant volume of the acid (15 mL) was carried out in a thermostatic cell using an air flow at 65 °C during 3 h. The oxidation proceeded in the conditions, the oxidation rate was independent of the air volume passing through the cell [82].

The concentrations of the additives were 1 mM compared with the initial concentration of LOOH in the reaction mixture. Oleic acid (1 mL), CHCl_3_ (12 mL), glacial AcOH (18 mL), and freshly prepared cold-saturated KI solution (1 mL) were placed in a flask and shaken for 2 min. Then distilled water (100 mL) and a 1% starch solution (1 mL) were added and the resulting mixture was immediately titrated with Na_2_S_2_O_3_ solution (0.01 N). Iodine released in an amount equivalent to that of LOOH was titrated with a standard thiosulfate solution. At the same time, a control test for reagents was carried out: all the reagents except for oleic acid were added to the flask. The LOOH concentration was calculated according to the following formula:

[LOOH] = [(V_s_ − V_c_) × 0.001269 × K × 100]/m, where vs. is the volume of 0.01 N Na_2_S_2_O_3_ solution, consumed during the titration of working sample, mL; V_c_ is the volume of 0.01 N Na_2_S_2_O_3_ solution, consumed during the titration of control, mL; K is the conversion factor to exactly 0.01 N Na_2_S_2_O_3_ solution; m is the mass of studied oleic acid; and 0.001269 is the amount of I_2_ expressed in g, equivalent to 1 mL of 0.01 N Na_2_S_2_O_3_ solution. The [LOOH, mmol/L] content equal to 1% corresponded to 78.7 mM of active O_2_ per 1 L of lipids.

The accumulation rate for the final oxidation products (TBARS) was determined according to a modified standard method [83]. Samples (0.01 mL) of oleic acid were taken from the thermostat every 30 min. They were introduced into a test tube containing a mixture of Tris buffer (0.8 mL), distilled water (1.2 mL), and freshly prepared thiobarbituric acid solution (0.8%, 1 mL). The tube was placed for 10 min in a boiling water bath, and after cooling, the absorption of the samples was measured in comparison with that of the control at λ = 532 nm. A similar mixture, but without added oleic acid, was used as the control. The concentration of carbonyl compounds was calculated according to the formula:

[TBARS] = (E × 3)/0.156, where [TBARS] is the content of carbonyl compounds, nM; E is the extinction of a sample relative to the control; 3 is the sample volume, mL; and 0.156 is the extinction of malondialdehyde (1 nmol) dissolved in 1 mL at λ = 532 nm.

### 3.11. Determination of TBARS Accumulation in the Russian Sturgeon Liver Homogenate

The experiments in vitro were carried out using the liver of a Russian sturgeon raised in a unique aqua complex for the reproduction of valuable fish species of the Federal Research Center Southern Scientific Center of the Russian Academy of Sciences within the framework of State Assignment (Project No. 122020100328-1). All manipulations were conducted according to the international rules GLP (Good Laboratory Practice). The samples of fish liver were fixed in the cold.

The LPO intensity was estimated according to the accumulation of carbonyl products forming a colored complex with thiobarbituric acid (TBARS) [83,84]. A liver of Russian sturgeon (10 g) was homogenized in the cold, and the studied compounds were added at a concentration of 0.1 mM in EtOH to a solution of 1.2% potassium chloride (390 mL) precooled to 0–4 °C. The absence of any influence of EtOH on the LPO rate in the control was preliminarily established under these conditions. The resulting mixture was poured into flasks and incubated with the added studied compounds at a temperature of 5 °C for 48 h, sampling the mixture (2 mL) at a certain time interval into tubes for subsequent centrifugation. Solutions of 2.6 mM ascorbic acid (0.1 mL), 0.04 mM Mohr’s salt (0.1 mL), and 40% trichloroacetic acid (1 mL) were added to these tubes. The tubes were placed in a water bath at 37 °C for 10 min and then centrifuged for 10 min (3000 rpm). The supernatants (2 mL) were placed into clean tubes, 0.8% thiobarbituric acid solutions (1 mL) were added, the samples were placed in a boiling water bath for 10 min, and then they were cooled in ice water down to room temperature (~20 °C). After cooling, CHCl_3_ (1.0 mL) was added to the samples to obtain a clear solution, and the resulting mixtures were centrifuged for 15 min (3000 rpm). The supernatant was sampled, and the extinction of sample was measured using a SF-103 spectrophotometer at λ = 532 nm relative to the control sample. The calculation was carried out according to the formula:

X = (E × 3 × 3.2)/(0.156 × 2), wherein X is the content of carbonyl products in the starting homogenate, nmol; E is the extinction of samples; 3 is the volume of samples, mL; 3.2 is the total volume of studied samples, mL; 0.156 is the extinction of malondialdehyde (1 nmol) dissolved in 1 mL at λ = 532 nm; 2 is the volume of supernatant used to determine carbonyl products, mL.

### 3.12. MTT-Test

Cell survival was determined by the MTT [3-(4,5-dimethyldiazol-2-yl)-2,5-diphenyl tetrazolium bromide] assay by Niks method with small modifications [85]. HCT-116 (colon carcinoma), MCF-7 (breast adenocarcinoma), A-549 (lung adenocarcinoma) and SW-480 (colorectal carcinoma) human cancer cell lines and WI-38 (cell line composed of fibroblasts) were cultured in DMEM (PanEco, Moscow, Russia) with Glutamin (PanEco, Moscow, Russia) and antibiotics (PanEco, Moscow, Russia) in CO_2_ (5%) at 37 °C. The compounds were dissolved (20 mM) in DMSO and then added to the cell culture medium at the required concentration with a maximum DMSO content of 0.5 *v/v*%. At these concentrations, DMSO had no effect on cell viability, as was shown in the control experiments. Cells were cultured in 96-well plates (7000 cells/well) and treated with various concentrations (from 0.01 to 100 mM) of the test compounds as well as doxorubicin at 37 °C for 72 h. Cell viability was determined using the MTT assay, which quantified dehydrogenase activity. Then, the cells were incubated at 37 °C for 50 min with a solution of MTT (10 mL, 5 mg × mL^−1^) (Sigma-Aldrich, St. Louis, MO, USA). The supernatant was discarded, and cells were dissolved in DMSO (100 mL). The optical density of the solution was measured at 570 nm with use of a multiwall-plate reader (Anthos Zenyth 2000rt, Biochrom, Cambridge, UK), and the percentage of surviving cells was calculated from the absorbance of untreated cells. Each experiment was repeated at least three times, and each concentration was tested in at least three replicates. Data were presented as a graph of the percentage of surviving cells versus the concentration of the test substances. The meanings of 50% inhibition concentration (IC_50_) with standard deviation were calculated using GraphPad Prism Version 5.03 for Windows (GraphPad Software, San Diego, CA, USA).

### 3.13. Annexin V/Dead Cell Assay and Cell Cycle Analysis

HCT-116 (colon carcinoma) and A-549 (lung carcinoma) cells (1 × 10^6^) were seeded in a six-well plate and were incubated for 24 h. After attachment of cells to the bottom of the plate, the medium was treated with the selected compound at doubled IC_50_ concentrations and incubated for 24 and 48 h. After incubation, the cells were harvested by trypsinization, precipitated by centrifugation (3000 rpm), washed with PBS, recentrifuged, and the buffer was removed. DMEM was added to achieve a cell concentration of 400–1000 cells/µL. The Muse Annexin V & Dead Cell Kit reagent (100 µL) was added to the cells and inhibited for 20 min at room temperature in the dark. The results were recorded on a Muse Cell Analyzer flow cytometer (Merck, Rahway, NJ, USA).

For the cell cycle analysis, HCT-116 cells (1 × 10^6^) were seeded in a six-well plate and were incubated for 24 h, and then the medium was treated with the compound at doubled IC_50_ concentrations and incubated for 24 h. After incubation, the cells were harvested by trypsinization, and precipitated by centrifugation (5000 rpm). After precipitation, the supernatant was removed, washed with PBS, centrifuged, fixed with 70% EtOH and incubated for at least 3 h at −20 °C. After incubation, 200 µL of the cell suspension was collected, centrifuged, the supernatant was removed, and washed with 200 µL of PBS. Then, the cells were stained with 200 µL of the Muse Cell Cycle reagent and incubated for 30 min at room Temperature in the dark. Cell cycle analysis was performed using a Muse Cell Analyzer flow cytometer.

### 3.14. Statistical Analysis

The acquired results were statistically processed using the Student *t*-test (implemented in Microsoft Excel software), and the average value and standard deviation were calculated. Between six and ten repeats were carried out for each experimental determination, and the nature of influence was estimated using the average values of activity, taking into account the experimental error (*p* < 0.01). Pearson’s correlation coefficient (r) was determined to establish the relationship between the values of the parameters of antioxidant activity and cytotoxicity of aromatic oligosulfides on various model systems.

## 4. Conclusions

In this work, we have explored the antioxidant and cytotoxic activities of aromatic oligosulfides. The probability of acting as a ROS scavenger and antidote was shown in silico for the studied compounds. Antioxidant activity was estimated by in vitro model reactions (FIC, CUPRAC, FRAP, NBT, DPPH-tests) and biochemical trials (LOX, SOD-activity, LPO). Only disulfides with phenolic groups demonstrated a radical scavenging activity in the DPPH-test, indicating their action mainly through the transfer of hydrogen atoms (HAT mechanism). Significant reducing activity in the CUPRAC and FRAP tests was also revealed for disulfides with phenolic fragments. Oligosulfides exhibited moderate iron-chelating activity, an order of magnitude lower than the EDTA standard. Derivatives with *HO*-groups showed Fe^2+^-chelating activity most effectively, while derivatives with *CH_3_O*-groups chelated to a lesser extent. The antiradical and SOD-protective activity of aromatic sulfides with respect to O_2_^−•^ generated in enzymatic and non-enzymatic model systems was revealed. The highest superoxide anion-radical scavenging activity was observed in the model system of quinoid oxidation of adrenaline in an alkaline carbonate buffer. There was no unambiguous regularity in the manifestation of activity in relation to O_2_^−•^ depending on the presence of *HO*-, *CH_3_O*-groups, disulfide or trisulfide linkers. The studied oligosulfides did not show anti-inflammatory activity. In the model system of oleic acid peroxidation, disulfides with phenolic fragments exhibited the highest inhibitory activity. This can be explained by the intramolecular synergism of the antiradical action of two phenol fragments and the antiperoxide action of sulfur (II) atoms. A pronounced prooxidant activity of diphenyl disulfide and the absence of anti-/prooxidant activity in methoxy derivatives of disulfides was established. Using the in silico method, the possibility of exhibiting anticancer activity for the studied compounds was predicted with a high probability. The antiproliferative status of aromatic oligosulfides on human cancer cell lines (SW-480, HCT-116, MCF-7, A-549) was assessed using the MTT test. IC_50_ values were in the range of 15.5–45.3 µM for all compounds. The maximum cytotoxicity was found for bis(2-methoxyphenyl) disulphide 2 and bis(3,4-dimethoxyphenyl) disulfide **3** on HCT-116 and A-549 cancer cells. The low cytotoxicity of bis(3,5-di-*tert*-butyl-4-hydroxyphenyl) disulphide was explained by the presence of the antioxidant 2,6-di-*tert*-butylphenol fragment in the structure and was consistent with the in silico prediction. The cell death mechanism study demonstrated that the bis(3,4-dimethoxyphenyl) disulfide **3** significantly induced apoptosis in HCT-116 cells after 48 h, and cell cycle arrest in HCT-116 cancer cells was recorded in the G2/M phase. There were significant and moderate negative correlations between IC_50_ values for cancer cell viability and the rate of LOOH accumulation in oleic acid (r = −0.7703 ÷ −0.9099). Positive correlations were established between the activities of CUPRAC, FRAP and Fe^2+^-chelating activity and between the cytotoxicity of the compounds and their antiradical activity towards O_2_^−•^. The obtained results can later be used to develop a strategy for the synthesis and study of target-oriented compounds with predictable biological activity.

Thus, it was found that the greatest antioxidant effect was characteristic of compounds with redox-active groups (HO-, CH_3_O-), and the number of sulfur atoms in the oligosulfide did not have a noticeable effect on the manifestation of both antioxidant activity and cytotoxicity. The presence of a spatially obstructed phenolic fragment reduced the antitumor effect of the compound, whereas the presence of several methoxy groups, on the contrary, enhanced it. Compound **3** did not exhibit a pronounced anti-/pro-oxidant effect, and is a promising leader compound for further research on the development of effective and safe chemotherapeutic drugs based on natural biologically active compounds.

## Figures and Tables

**Figure 1 molecules-27-03961-f001:**
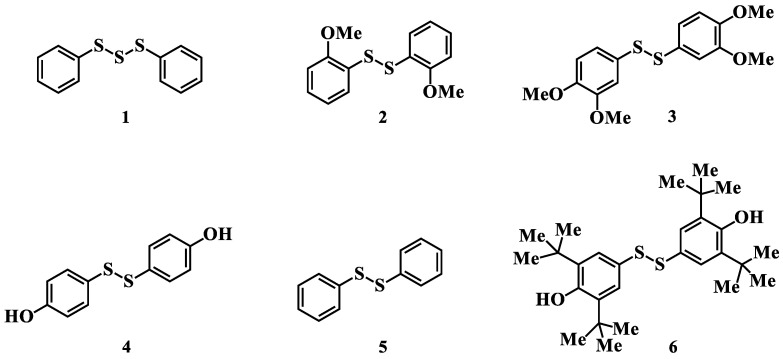
Structures of organosulfur compounds **1**–**6**: diphenyl trisulfide (**1**), bis(2-methoxyphenyl) disulfide (**2**), bis(3,4-dimethoxyphenyl) disulfide (**3**), 4,4′-dihydroxydiphenyl disulfide (**4**), diphenyl disulfide (**5**), bis(3,5-di-*tert*-butyl-4-hydroxyphenyl) disulfide (**6**).

**Figure 2 molecules-27-03961-f002:**
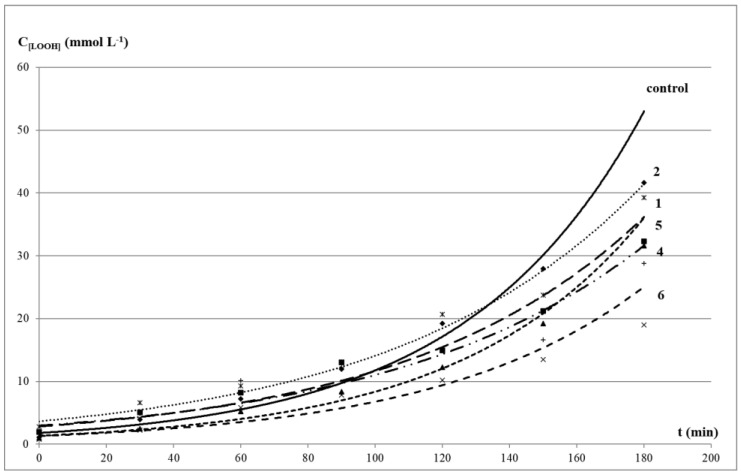
Kinetic curves of LOOH formation in the presence of 1 mmol/L additives at 65 °C: (**control**) oleic acid without additives; (**1**) diphenyl trisulfide; (**2**) bis(2-methoxyphenyl) disulfide; (**4**) 4,4′-dihydroxydiphenyl disulphide; (**5**) diphenyl disulfide; (**6**) bis(3,5-di-*tert*-butyl-4-hydroxyphenyl) disulphide.

**Figure 3 molecules-27-03961-f003:**
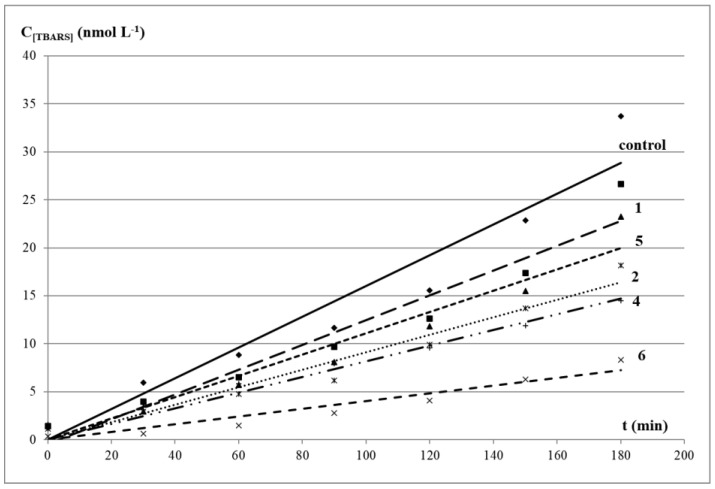
Kinetic curves for the accumulation of TBARS in oleic acid in the presence of 1 mmol/L additives at 65 °C: (**control**) oleic acid without additives; (**1**) diphenyl trisulfide; (**2**) bis(2-methoxyphenyl) disulfide; (**4**) 4,4′-dihydroxydiphenyl disulphide; (**5**) diphenyl disulfide; (**6**) bis(3,5-di-*tert*-butyl-4-hydroxyphenyl) disulphide.

**Figure 4 molecules-27-03961-f004:**
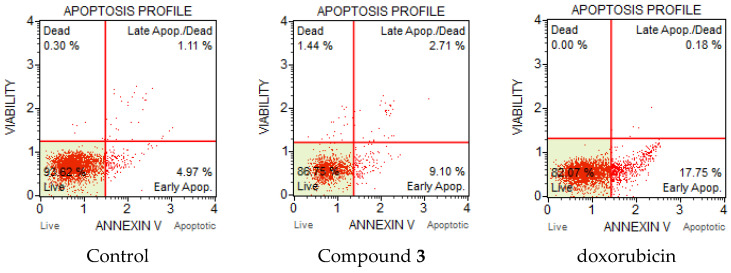
Apoptotic profile of HCT-116 cancer cells after treatment with compound **3** and doxorubicin at 2 × IC_50_ (μM) concentration after 24 h.

**Figure 5 molecules-27-03961-f005:**
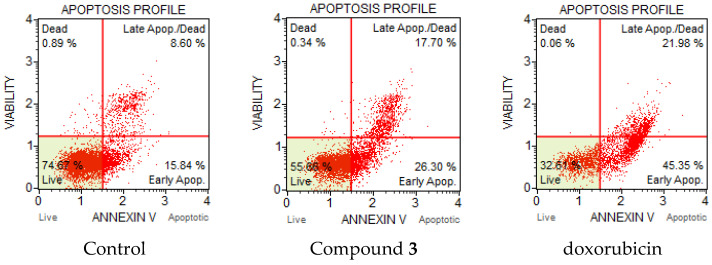
Apoptotic profile of HCT-116 cancer cells after treatment with compound **3** and doxorubicin at 2 × IC_50_ (μM) concentration after 48 h.

**Figure 6 molecules-27-03961-f006:**
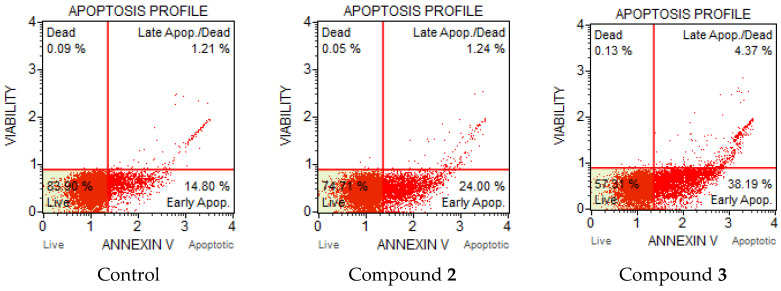
Apoptotic profile of A-549 cancer cells after treatment with compounds **2** and **3** at μM after 48 h. Concentration of compounds 2 × IC_50_.

**Figure 7 molecules-27-03961-f007:**
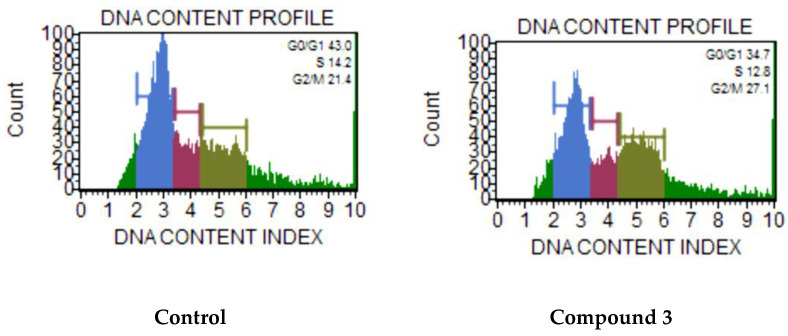
Cell cycle of HCT-116 cancer cells after treatment with compound **3** at μM after 24 h. Concentration of compounds 2 × IC_50_.

**Table 1 molecules-27-03961-t001:** Antioxidant activity and cytotoxicity prediction for compounds **1**–**6**.

Activity		Compounds					
		1	2	3	4	5 [26]	6 [26]
Oxygen scavenger	P*_a_*	0.590	0.568	0.558	0.605	0.621	0.519
	P*_i_*	0.025	0.030	0.033	0.022	0.018	0.044
Nitric oxide scavenger	P*_a_*	0.260	0.285	0.271	0.349	0.274	0.278
	P*_i_*	0.017	0.010	0.013	0.004	0.013	0.012
Free radical scavenger	P*_a_*	0.208	0.359	0.330	0.368	0.230	0.473
	P*_i_*	0.070	0.021	0.025	0.020	0.056	0.012
Antioxidant	P*_a_*		0.143	0.174	0.280		0.549
	P*_i_*		0.111	0.074	0.027		0.005
Antidote	P*_a_*	0.205	0.192	0.169	0.285	0.228	0.317
	P*_i_*	0.101	0.117	0.145	0.039	0.078	0.027
**Cytotoxicity**							
MCF-7	P*_a_*	0.936	0.707	0.859	0.799	0.882	0.403
	P*_i_*	0.005	0.019	0.006	0.011	0.005	0.082

P*_a_*—probability of presence of biological activity, P*_i_*—probability of lack of biological activity.

**Table 2 molecules-27-03961-t002:** Antioxidant and Fe^2+^-chelating activity of compounds **1**–**6**.

Compounds	DPPH, %	TEAC_CUPRAC_	TEAC_FRAP_	FIC, % of Inhibition	FIC, IC_50_ mM
**1**	non active	non active	0.06 ± 0.01	25.8 ± 1.1	7.76 ± 0.02
**2**	non active	non active	0.58 ± 0.05	7.1 ± 0.2	>10
**3**	non active	0.06 ± 0.01		7.5 ± 0.3	>10
**4**	13.42 ± 0.02	1.80 ± 0.06	0.69 ± 0.07	47.2 ± 1.5	5.62 ± 0.04
**5**	non active	0.10 ± 0.01 *	0.06 ± 0.05	73.7 ± 2.1 *	3.74 ± 0.02
**6**	81.56 ± 0.07	2.04 ± 0.05 *	0.69 ± 0.02	28.9 ± 1.3 *	6.10 ± 0.01

* [26].

**Table 3 molecules-27-03961-t003:** Antiradical capacity indicators of compounds **1**–**6**.

Compounds	NBT, % Inhibition	Antiradical Activity, % Inhibition	SOD Activity of Liver Sturgeon, % Inhibition	LOOH, k_0_/k_1_	TBARS, % Inhibition
**1**	non active	30 ± 1.9	41 ± 2.7	0.78 ± 0.15	21.0 ± 0.18
**2**	non active	50 ± 3.1	60 ± 3.3	0.86 ± 0.11	42.9 ± 0.76
**3**	24.3 ± 0.05				
**4**	57.8 ± 0.07	36 ± 2.3	75 ± 5.6	0.61 ± 0.03	39.0 ± 0.49
**5**	non active	56 ± 2.9 *	61 ± 4.8 *	0.79 ± 0.11 **	24.6 ± 0.29 **
**6**	non active	37 ± 4.8 *	85 ± 3.0 *	0.49 ± 0.05 **	73.1 ± 1.30 **

* [26]; ** [29].

**Table 4 molecules-27-03961-t004:** Results of the MTT assay on different cell lines.

Compound	IC_50_, μM				
	MCF-7	SW-480	A-549	HCT-116	WI-38
**1**	37.6 ±3.7	23.8 ± 4.5	43.6 ± 10.1	18.8 ± 1.4	52.7 ± 8.5
**2**	30.5 ± 5.2	22.4 ± 3.8	29.6 ± 7.5	24.4 ± 2.3	31.9 ± 5.2
**3**	37.7 ± 2.7	21.7 ± 2.4	15.5 ± 3.4	18.9 ± 2.1	32.8 ± 5.4
**4**	45.3 ± 3.1	39.4 ± 7.9	29.7 ± 6.1	22.0 ± 1.5	28.7 ± 5.3
**5**	34.8 ± 12.2	30.6 ± 5.6	39.2 ± 8.7	22.1 ± 1.7	33.7 ± 9.1
**6**	>50	>50	>50	>50	115 ± 75
cisplatin	>30	17.6 ± 5.4	20.1 ± 5.6	9.1 ± 2.3	
doxorubicine	0.28 ± 0.03	0.45 ± 0.06	0.24± 0.02	0.65 ± 0.05	

MTT = 3-(4,5-dimethylthiazol-2-yl)-2,5-diphenyl tetrazolium bromide.

**Table 5 molecules-27-03961-t005:** Pearson’s correlation coefficient between the values of the parameters of antioxidant and cytotoxic activity on various model systems in vitro.

	Antiradical Activity O_2_^–•^	SOD-Protective Activity	LOOH in Oleic Acid	CUPRAC	FIC	FRAP	MCF-7	SW-480	A-549
SOD-protective activity	−0.5725								
LOOH in oleic acid	−0.4721	0.4041							
CUPRAC	0.3379	0.0013	−0.9380						
FIC	−0.1243	−0.2462	−0.7742	0.9595					
FRAP	0.5348	0.2016	−0.5724	0.9935	0.1910				
MCF-7	0.7123	−0.6863	−0.9076	0.7481	0.5901	0.5062			
SW-480	0.7566	−0.6695	−0.9099	0.7716	0.5959	0.5287	0.9884		
A-549	0.7188	−0.8636	−0.7703	0.6538	0.6323	0.2823	0.9225	0.9350	
HCT-116	0.8338	−0.7431	−0.8063	0.6635	0.4731	0.4977	0.9760	0.9769	0.9444

LOOH in oleic acid: k_0_ and k_i_—the reaction rate constant of the pseudo-first order of accumulation of LOOH in the control experiment.

## Data Availability

The data presented in this study are available in this article.

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
