# Peer review of "Antioxidant Activity and Cytotoxicity of Aromatic Oligosulfides"

_molecules, 2022, doi:10.3390/molecules27123961_

Round 1

Reviewer 1 Report

This work studied the antioxidant and cytotoxic activities of aromatic oligosulfides, showing different ositive correlations  between the activities of CUPRAC, FRAP and iron chelating activity and between  the cytotoxicity of the compounds and their antiradical activity . These results may be used ibn future to synthesize and study compounds with predictable biological activity. 

I found some english periods hard to comprehend, so probably an english revisions should be performed before considering this paper for publication

Thank You

Author Response

We corrected English and we hope that we could eliminated all errors. We apologize for these errors. We made corresponding changes marked by blue in the text (both in the manuscript and supporting materials).

Reviewer 2 Report

In the sentence of lines 21-23 a keyword is missing. 

 The highest cytotoxicity on cell lines A-549 and HCT-116 was found for  bis(3,4-dimethoxyphenyl) disulfide 3 a significant induction of apoptosis in HCT-116 cells by it’s (what?) indicates the promise of using this disulfide as an antitumor agent. 

The sentence in lines 140-142 does not read right. Please rephrase.

It is known that depending on the 140 structure of the antioxidant and conditions can dominate one of the mechanisms of the 141 antiradical action of antioxidants: 

Line 147 replace were with was

Line 160 give full name of acronym CUPRAC

Lines 173-174 delete re-action

Line 226 give full name of acronym NBT

Line 240 replace disulfide 1-6 with disulfides 1-6

Line 243 Superoxide dismutase (SOD) presents in almost all aerobic cells...change to Superoxide dismutase (SOD) is present in almost all aerobic cells...

The sentence in lines 255-260 is too long. Please split and rephrase.

Line 281 give full name of acronym LOX

Line 282 give full names of acronyms LOOH and TBARS

Legend of Figure 2 in lines 305-308 justify left

Line 394 give full name of BisPMB

Insert blank line between lines 509 and 510 , 519 and 520, 525 and 526

In Materials and and Methods Section insert in  all 2C6H5,  2C6H4, 2CH3O, 2C6H3 etc 2 x  ...

Line 808 disulfide 3 ( insert compound number)

Author Response

We thank you for their work with the manuscript and comments, which helped to improve it. We agree with the comments and made corresponding changes marked by blue in the text (both in the manuscript and supporting materials). Please find attached detailed answers to comments in separate file.

Reviewer 3 Report

The authors report a paper that is devoted to in silico and in vitro evaluation of anti-oxidant and cytotoxic properties of synthetic analogs of natural compounds – aromatic oligo sulfides. My first impression is very good. The authors comment very well and comprehensively on the experiments they have chosen and the obtained results are very well described.

However, I have some questions and recommendations.

Please unify the structural formulas in figure 1. Some of the bonds are shorter than others. Please for compound 6 draw the t-butyl residues.

In table 1, as the cytotoxicity and antidote prediction results are included, please remove the bold Antioxidant word. Just leave Activity (after row 132)

Please check one more time your spectral data. For compound 1 you report the NMR data as singlet (s) peaks (Row 515-518). The article published in the literature (Zysman-Colman and Harpp doi: 10.1021/jo0265481)report the same compound and describe the signals as multiples (m). If it is the same compound, why there is a difference? Also, I suggest the authors cite this article. 

I recommend the article be published after minor corrections.

Author Response

(The authors gave the same response as above.)

Reviewer 4 Report

Dear authors,

There is some incorrect English expression in the main text and authors need to check carefully for this before submitting a revised version.

Also, several matters posed in the paper need to be addressed:

1)    Introduction: This chapter could be improved to enhance the current research topic. The focus objective of this research work should be expanded and clearly stated as it is not clear. What is the novelty of this research project?

2)    Chapter 3: Materials and Methods: Overall, it is too lengthy. It is recommended to summarize each experimental procedure.

3)    Section 3.1: The arrangement is a little confusing. The characterization of the prepared compounds could be arranged and reported together instead of separately at the moment. Also, as compounds 1,2,3 are known, what is the comparison of the compounds with literature.

4)    Conclusion need to be improved to highlight and show the significance of the research work.

5)    References: There were some incorrect formatting for some references. Please do recheck.

Author Response

(The authors gave the same response as above.)

Round 2

Reviewer 1 Report

The paper improved after one round of revisions. It is in my opinion eligible to be published